# The Importance of Assay Imprecision near the Screen Cutoff for Newborn Screening of Lysosomal Storage Diseases

**DOI:** 10.3390/ijns5020017

**Published:** 2019-03-27

**Authors:** Bruce H. Robinson, Michael H. Gelb

**Affiliations:** Department of Chemistry, University of Washington, Seattle, WA 98195, USA

**Keywords:** newborn screening, cutoff values, assay imprecision, false positives, false negatives, screening assays

## Abstract

For newborn screening (NBS) of lysosomal storage diseases, programs measure enzymatic activities in dried blood spots (DBS) and, in most cases, act on samples where the measurement is below a specific cutoff value. The rate of false positives and negatives in any NBS program is of critical importance. The measured values across a population of newborns are governed by many factors, and in this article we focus on assay imprecision. Assay parameters including the Analytical Range and the Z-Factor have been discussed as a way to compare assay performance for NBS of lysosomal storage diseases. Here we show that these parameters are not rigorously connected to the rate of false positives and negatives. Rather, it is the assay imprecision near the screen cutoff that is the most important parameter that determines the rate of false positives and negatives. We develop the theoretical treatment of assay imprecision and how it is linked to screen performance. What emerges is a useful type of parametric plot that allows for rigorous assessment of the effect of assay imprecision on the rate of false positives and false negatives that is independent of the choice of screen cutoff value. Such plots are useful in choosing cutoff values. They also show that a high assay imprecision cannot be overcome by changing the cutoff value or by use of postanalysis, statistical tools. Given the importance of assay imprecision near the cutoff, we propose that quality control DBS are most useful if they span a range of analyte values near the cutoff. Our treatment is also appropriate for comparing the performance of multiple assay platforms that each measure the same quantity (i.e., the enzymatic activity in DBS). The analysis shows that it is always best to use the assay platform that gives the lowest imprecision near the cutoff.

## 1. Introduction

In most newborn screening (NBS) programs, measurement is made on analytes or enzymatic activities present in dried blood spots (DBS) on NBS cards. In cases where a low value of the measurement is characteristic of a disease, for example the activity of an enzyme, the NBS program sets a cutoff value such that newborns displaying an assay value below the cutoff are considered screen-positive in the first-tier analysis. Often, additional second-tier tests may be carried out before the NBS laboratory reports the result as screen-positive. For other screens, disease status is suggested if the measured analyte is above the cutoff value (for example biomarkers that are elevated due to the absence of an enzyme). Each NBS laboratory decides on a method for setting cutoff values. Usually this is done by carrying out the assay on a set of DBS from patients confirmed to have the disease and those who are healthy. NBS laboratories often adjust their cutoff values over time to reduce the number of false positives and false negatives.

It is useful to consider features of the NBS assay that affect the number of false positives and false negatives as false positives are burdensome for the laboratory, and false negatives are to be avoided. In this article, we report the results of a rigorous analysis of the false positive/false negative problem using standard statistical analysis adapted to data representative of that obtained in NBS laboratories.

## 2. Population Distributions of Assay Measurements and Definition of False Positives and False Negatives

In Figure 1A we show the hypothetical distributions of the NBS assay parameter. These plots are typically done by binning, whereby one carries out an assay on a large set of newborn DBS (say 100,000), counts the number of newborns who display a certain small span of assay values (say 1.0–1.1 μmol/h/L for an enzymatic assay), and divides this number by the total number of newborns to obtain the fraction of newborns in this assay bin. These histograms are typically found to be well-fitted by a log-normal mathematical function as shown in Figure 1A. The reason for use of the Log-normal function is described below. Distributions of this type are called probability distribution functions (PDFs). They are normalized so that the sum of probabilities in all of the bins (or the area under the curve) is unity.

In this publication, we consider the case that newborns with the disease display a low value of the assay parameter compared to those that do not have the disease (typical of a NBS assay based on enzymatic activity). The arguments presented here apply equal well to the opposite case where a high assay value is typical of the disease case (for example an elevated biomarker due to a deficient enzyme). In NBS we are interested in the PDF for a set of DBS from a population of healthy newborns and from a collection of newborns with the disease of interest. Two such hypothetical distributions are shown in Figure 1B, where it is seen that they partially overlap. This overlap region is shown as an expanded plot (Figure 1C). Each NBS laboratory chooses a cutoff value in order to see if any particular newborn is found to be screen-positive (enzymatic activity below the cutoff) or screen-negative (enzymatic activity above the cutoff). The fraction of healthy newborns that are false positives is defined as the area under the healthy newborn PDF to the left of the cutoff, and the total number of false positives is this fraction times the total number of healthy newborns screened (Figure 1C). The fraction of newborns with the disease that are false negatives is the area under the disease newborn PDF to the right of the cutoff, and the total number of false negatives is this fraction times the total number of disease newborns (Figure 1C). For a rare disease, if say 100,000 random newborns are tested, the number of false positives expected is nearly equal to the fraction of false positives times 100,000 since almost all of the 100,000 are healthy. The total number of false negatives expected out of 100,000 can only be stated if the disease frequency is known so that the number of expected true disease newborns can be estimated. For example, if the disease frequency is assumed to be five per 100,000 and the fraction of false negatives is 0.2, one expects to find four true positives out of 100,000 newborns screened and to miss one true positive (which is misidentified as a false negative).

Assay of analytes in samples taken from large populations tend to display a Normal (also known as a Gaussian) distribution. For example, it would be surprising to see a flat distribution of assay values with near-vertical drop-off at the low and high ends. Enzyme activity values measured in a population have to be real positive numbers, and so the left tail of the distribution goes to the origin and stops. The right tail on the other hand has no such constraint. This leads to a skewing of the normal curve that is more obvious as the population mean assay value is less than ~5-fold larger than the distribution width. The log-normal distribution allows for this skewing, and the observed data is usually well modeled by this function (given in Appendix A). There is no deep underlying reason to explain this, but rather the log-normal function provides a convenient, continuous, and well-behaved function for statistical analysis, which is consistent with the described constraints on the data. Strictly speaking, when the enzymatic activity is close to zero and with finite imprecision in the measurement, the activity measured may actually be less than that for the no-enzyme blank. In this case one obtains an enzymatic activity less than zero if the assay values are blank corrected. Thus, the PDF does not have to pass through the origin of Figure 1B, but it will pass close enough to the origin to lead to the skewing, and the log-normal function is still appropriate.

False positives are sometimes classified as pseudodeficiencies if the DNA sequence of the relevant gene shows a variation that is the likely reason for the decrease in enzymatic activity that has not reached a sufficiently low level to cause the disease. Depending on the amount of residual enzymatic activity (and possibly other factors), diseases can be early onset with severe symptoms or later onset with less severe symptoms. In this chapter we do not consider these variations; rather we speak only of unaffected and affected patients.

## 3. Lysosomal Storage Diseases as an Example

The current thinking is that a lysosomal storage disease results when a lysosomal protein, usually an enzyme or a transporter, becomes dysfunctional (due to mutations in its gene) to the point that a metabolite, usually the substrate for the enzyme or transporter, builds up to an abnormal level leading to cell death and associated tissue pathology. All NBS of lysosomal storage diseases currently performed are based on measurement of the amount of lysosomal enzymatic activity in a constant area of a DBS on newborn screening cards (typically a 3-mm punch) [1]. There are possible reasons that may complicate the relationship between the measured enzymatic activity and the disease status. Examples include additional genetic factors besides mutations in the relevant lysosomal enzyme gene, differential white cell count in the DBS, differences in handling and storage of DBS, and imprecision in the measurement of the enzymatic activity. In this paper, we focus only on the latter element because we are interested in the general problem of comparing multiple assay platforms, all of which measure the amount of enzymatic activity in a 3-mm DBS punch. All factors other than assay imprecision are common to a comparison of different assay platforms that all make use of the same DBS samples.

To verify that the enzymatic activity is proportional to the actual amount of enzymatic activity in the DBS, the Centers for Disease Control and Prevention (CDC) provides four standards called BASE, LOW, MEDIUM, and HIGH. The CDC BASE DBS is 0% whole blood/100% enzyme-depleted blood, LOW is 5% whole blood, MEDIUM is 50% whole blood, and HIGH is 100% whole blood [2]. Linearity between the measured enzymatic activity and the percent of whole blood in the DBS demonstrates that the assay response is proportional to the amount of enzyme in the DBS.

## 4. Assay Imprecision

The CDC uses the same Quality Control DBS to establish assay imprecision by measuring the enzymatic activity in twenty 3 mm punches from each of the 4 standards. Data is provided as Certificate Reports (https://www.cdc.gov/labstandards/nsqap_resources.html). The CDC provides the mean assay value for each Quality Control DBS as well as the 95% upper and lower confidence intervals. From the confidence intervals one obtains the standard error (also known as the standard deviation) in the measurement, denoted *σ_M_*, using the standard formula given below (found in most statistics textbooks):
*σ_M_* = (95% upper confidence limit—mean)/1.96 = (mean—95% lower confidence limit)/1.96

Strictly speaking, this type of imprecision measurement is due to the variation of the assay itself and due to the variation in the amount of enzyme in different punches from the same DBS. However, data shown in the accompanying paper proves that the assay variation is much larger than the punch variation (Figure 1 of Gelb et al. [1]), and thus we take the CDC-measured variation as the imprecision intrinsic to the assay.

## 5. Analytical Range

In general, a robust assay is one in which the response for the assay containing the enzyme is much higher than that of the blank that lacks enzyme. In this context, we define the analytical range as the assay response measured with a complete assay on a quality control HIGH DBS (typical of the maximal activity seen in a population) divided by the assay response due to all enzyme-independent events (i.e., the blank). The blank includes: (1) assay response due to product in the substrate as a contaminant; (2) assay response due to the substrate itself (for example some fluorogenic substrates have intrinsic fluorescence even if they are free of product [3]; mass spectrometry assays can sometimes give rise to in-source breakdown of substrate to product [3]); (3) assay response due to the sample matrix (i.e., blood); (4) assay response due to product formation not catalyzed by enzyme (spontaneous chemical decomposition of substrate). The contribution of each of these factors to the blank assay response can usually be determined with the proper control experiments.

In general, assays with low analytical ranges are not very reliable because of experimental error in the measurement of enzymatic activities. For example, if the assay gives a complete assay response of 100 (arbitrary units) and a blank response of 90, the analytical range is only 1.11, and any reasonable error in the blank will render the assay almost useless. Thus, reliable assays tend to have high Analytical Ranges, but as discussed below, the analytical range is not the most important parameter affecting the rate of false positives and false negatives in NBS assays.

## 6. Z-Factor

Z-factors include the Analytical Range as well as the error of the measurement. There is no single definition of the Z-factor as will be shown in this section. For example, consider the Z-factor equation below (equation 1).
Z = 1 − 3 × (*σ_N_* + *σ_D_*)/|(*µ_N_* − *µ_D_*)|(1)
In this equation *µ_N_* and *σ_N_* are the mean and standard deviation, respectively, for the enzymatic activity measured on a collection of DBS from normal patients, respectively, and *μ_D_* and *σ_D_* are measured on DBS from patients with the disease. Z approaches a maximum of 1 when the sum of the standard deviations (*σ_N_* + *σ_D_*) is much smaller than the difference in the means (*µ_N_* − *µ_D_*). In this case, the enzymatic activities of the disease and normal groups of people are well separated compared to the range of values for each group, and one can assign disease versus normal status to any new enzymatic activity value of a patient with high confidence. When Z = 0, this corresponds to *µ_N_* and *µ_D_* being separated by 3 × (*σ_N_* + *σ_D_*). Thus, the factor of 3 in equation (1) is an arbitrary constant.

The Z-factor (as well as the analytical range) is not a statistically rigorous formalism. The statistical formalism that is relevant to NBS studies described in this chapter is called the Student *t*-test. Applied here, one can construct a Student’s *t*-test to estimate if a certain measurement, say the enzymatic activity of a single newborn, belongs to the disease group or to the healthy group. The Student’s *t*-test leads to the assignment of a specific measurement to the two groups with a certain confidence. For example, one might find that a single newborn gives a measurement that puts this individual into the healthy group with 95% confidence and into the disease group with 5% confidence. Student’s *t*-tests are useful in setting the screen cutoff because one can define the confidence of assigning a measurement to the healthy group versus the affected group. One can explore how these confidences change with cutoff value.

One issue with Z-factors is they do not give a numerical sense of how good the test is. It has already been noted that Z-factor equations have arbitrary constants in them such as the factor of 3 in equation (1) above. The Student’s *t*-test on the other hand allows you to test the hypothesis that any particular newborn with a given NBS measurement belongs to the healthy group, or to the affected group, and to give the confidence (probability) of the group assignment. It is probably true in most cases that one test is better than the other if it has a higher Z-Factor, but again, the value of the Z-factor itself does not test the hypothesis of whether a particular newborn belongs to the healthy or to the affected group, which is the goal of all NBS programs.

While the Student’s *t*-test is statistically valid and has value, we do not discuss it further here. As stated above, the Student’s *t*-test is useful in choosing a screen cutoff, but in this chapter, we focus on assay parameters that affect the rate of false positives and false negatives.

## 7. Assay Imprecision Near the Screen Cutoff

Having shown that the analytical range and Z-factors are not particularly useful assay performance factors governing the rate of screen-positives and screen-negatives, we turn to other factors. Consider the situation in which there is finite assay imprecision near the screen cutoff. Then, a newborn may display a measured assay value just below the cutoff but actually has a true assay value (value measured if the imprecision were zero) above the cutoff, resulting in a probable false positive. Likewise, a newborn may display an enzymatic assay value just above the cutoff but actually has a true assay value below the cutoff, resulting in a probable false negative.

The mathematical treatment of PDFs that contain multiple contributions to the overall observed PDF is well founded. Consider two PDFs: The first is the PDF of assay values if the assay imprecision were zero. We refer to this is as the no-imprecision PDF. Note that it contains all variation in the population analysis due to variations to the DNA sequence of the relevant enzyme gene, variations in leukocyte count in blood, variation in enzyme level due to handing and preparation of DBS, and possibly other factors (discussed above), but it lacks the imprecision due to the assay itself. The second PDF is the imprecision due to the assay itself (imprecision PDF). The observed PDF (Figure 1) contains the no-imprecision PDF and the imprecision PDF blended together. The mathematics of blending two PDFs together is called convolution. The relevant equation for carrying out the convolution is given in Appendix A along with an explanation of the underlying process.

The amount of imprecision of the assay, *σ_M_*, in general, depends on the mean enzymatic activity in the DBS. To carry out the convolution, we need these values of *σ_M_*. In Figure 2 we show the imprecision data measured by the CDC on the four types of quality control DBS described above. This set is for the measurement of the enzymatic activity by tandem mass spectrometry due to α-glucosidase that is relevant to NBS of Pompe disease (data is available online at https://www.cdc.gov/labstandards/nsqap_resources.html). In Figure 2 we plot the four values of *σ_M_* versus the mean activity *μ_M_*. Note that *σ_M_* and *μ_M_* are *estimates* of the true parameters; the estimates are more accurate as the number of measurements in the imprecision analysis is increased. One can thus estimate the errors in *σ_M_* and *μ_M_*, and, according to standard textbooks in statistics, these both have a values of *σ_M_*/*N*^1/2^, where *N* is the number of punches analyzed. These standard deviations are shown as error bars in Figure 2. It is clear from this figure that *σ_M_* is not constant for all values of enzymatic activity measured in DBS, and thus a single value of *σ_M_* cannot be used for the convolution. To analyze the effect of imprecision properly, we need a continuous mathematical function that relates *σ_M_* to *μ_M_*. One such function is shown by the solid line through the CDC data in Figure 2 (the explicit function is given in the figure legend, its identity is not important).

In comparing two assay platforms, it is not the absolute values of the imprecision, *σ_M_*, that matters but the relative imprecision. Suppose assay platform 1 gives a value of 2 μmol/h/L for a DBS punch and the other platform gives 4 μmol/h/L for the same sample. This difference is due to the different substrates and buffer conditions used in each enzymatic activity assay. If *σ_M_* for platform 1 is found to be 0.5 μmol/h/L, an equivalent imprecision for platform 2 would be twice this value or *σ_M_* = 0.5 μmol/h/L. In this case, both platforms are equally imprecise. Strictly speaking, adjustment by scaling of this type works only if there are no offsets in the enzymatic activities in comparing one platform to another. For example, for measurement of GAA for NBS of Pompe disease, there is a different degree of interference from an off-target enzyme in fluorimetric versus tandem mass spectrometry assays [4]. This differential offset must be removed from observed enzymatic activity values before a simple proportional adjustment to *σ_M_* is made.

Using the imprecision data in Figure 2, we can carry out the convolution analysis (methodology is given in Appendix A). The observed PDF (Figure 1) and the imprecision data (Figure 2) are used to obtain the no-imprecision PDF for the affected and nonaffected newborns. With the latter in hand, the false positive and false negatives rates are obtained by integration of the appropriate regions of these PDFs using the definitions in Figure 1C and the integration formulas in Appendix A. The rates of false positives and false negatives depends on the screen cutoff (Figure 1C), and we compute all pairs of rates as the cutoff is varied. These pairs are shown as the black curve in Figure 3A closes to the origin. Again, this is for the case of no assay imprecision. The computer code for carrying out these analyses is available from the authors upon request, and the authors are also able to analyze new datasets upon request.

With the no-imprecision PDF in hand we compute the new PDF with different degrees of imprecision in the assay measurement, and again compute the false negative/false positive pairs for all cutoff values. As the first example, we consider the case of a uniform *σ_M_* = 0.5 μmol/h/L for all enzymatic activities. This leads to the black curve in Figure 3A furthest from the origin. The black curve passing close to the origin is for *σ_M_* = 0 μmol/h/L for all enzymatic activities. The red diamonds on each curve corresponds to a cutoff of 2.0 μmol/h/L. Thus, at this fixed cutoff the false positive and false negative rates are higher when assay imprecision is present. In statistical analysis, plots of the type shown in Figure 3 are a form of “receiver operating characteristic” curve (ROC curve, see Appendix A for further definitions).

Next we added imprecision to the assay in certain windows of enzymatic activity values. If the imprecision is high at assay values of 3 μmol/h/L or greater (*σ_M_* = 0.5 μmol/h/L) but relatively low at *σ_M_* = 0.1 μmol/h/L in the 0–3 μmol/h/L region, we obtain the blue curve in Figure 3A. This is almost identical to the no-imprecision curve (black curve) showing that imprecision well above the cutoff has only a small effect to increase the false positive rate and essentially no effect on the false negative rate (based on the position of the red diamonds). On the other hand, with *σ_M_* = 0.5 μmol/h/L well below the cutoff in the 0–1 μmol/h/L range with *σ_M_* = 0.1 μmol/h/L at 1 μmol/h/L or higher enzymatic activity, one obtains the red curve in Figure 3A. In this case the false positive rate changes very little, but the false negative rate increases substantially. This is the expected result since focusing the imprecision well below the cutoff shifts the PDF of the sick newborns more than that of the healthy newborns.

In Figure 3B we show an analysis where the zone of lowered imprecision (0.25 μmol/h/L) is swept through regions of assay values; the range of this zone is kept constant at 2 μmol/h/L, otherwise the imprecision is higher at 0.5 μmol/h/L. In this way we can assess the impact of less imprecision below, at and above our nominally chosen cutoff of 2 μmol/h/L. The two black curves in Figure 3B, identical to those in Figure 3A, represent no-imprecision and full imprecision references. The blue curve is the case where the region of smaller *σ_M_* is centered at 1 μmol/h/L (i.e., covers the range of 0 to 2 μmol/h/L). The green curve has the smaller imprecision centered at the cutoff of 2 μmol/h/L (i.e., 1–3 μmol/h/L). The red curve has the smaller imprecision centered at 3 μmol/h/L (i.e., 2–4 μmol/h/L). Finally, the yellow curve has the smaller imprecision centered at 4 μmol/h/L (i.e., 3–5 μmol/h/L). The diamonds on the curves in Figure 3B correspond to an assay value of 2.0 μmol/h/L, and, at this assay value, which can be considered a cutoff, the green and red curves show the smallest false positive rate. This shows that minimizing the imprecision just above the cutoff is most effective at reducing the false positive rate. Similarly, the green and blue curves show the smallest false negative rate. Note that the green curve shows improvement in both rates, although each to a lesser extent. This analysis leads to the most important conclusion of this study, which is that imprecision near the cutoff affects both the false positive and false negatives rates to the largest extent.

Since the imprecision of the assay platform is intrinsic to the platform, the only improvement in imprecision obtainable by the NBS laboratory is to repeat the DBS measurement *N* times, since the error drops as *N* increases according to the formula
(2)$$\frac{\sigma_{M}}{\sqrt{N}}$$


Thus, repeating sample measurements four times would decrease the imprecision by a factor of 2, as was carried out in the analysis shown in Figure 3B. Figure 3B indicates that repeat measurement is best carried out for the borderline cases around the cutoff, especially just above the cutoff.

The curves shown in Figure 3B allow us to pick a new cutoff and retesting window which will minimize either both false rates or prioritize one over the other. For example, if the goal is to simply minimize false positives (without impacting the false negative rate), then one would choose a cutoff on the green curve corresponding to the lowest false positives, which would be a cutoff of ~1.6 μmol/h/L, with retesting on assay values in the range of 1 to 3 μmol/h/L. Or, as another example, if the goal is to minimize both false rates, one would choose a cutoff of 1.8 μmol/h/L, with retesting in the range of 0–2 μmol/h/L (the blue curve). Note that in both of these examples, retesting should occur right about the new cutoff. Carrying out an analysis of samples to mimic the blue curve in Figure 3B in practice is more challenging because of the inherent error in the individual measurements. To be sure to find samples that have half the error in the 0 to 2 μmol/h/L range, samples that initially test in the 0 to 3 μmol/h/L range must be retested to capture 99% of those that belong in the 0 to 2 μmol/h/L range. Again, this shows that retesting still should occur near the cutoff.

## 8. Comparison of Different NBS Assay Platforms

As mentioned already, the statistical analysis developed in this article is useful to compare the NBS assay performance of different platforms that aim to measure some biomarker, i.e., the amount of enzymatic activity in DBS. It is reasonable to propose that the most accurate comparison starts with a single population of newborns, and for each newborn, multiple punches are taken from a single DBS with each punch submitted to a different assay platform. In this way, false positive and false negative rates are not influenced by different study populations.

The number of false positives and false negatives measured with any given assay platform depends on the cutoff used for each platform (Figure 1). Thus, any comparison of the rate of false positives and false negatives obtained with multiple platforms requires careful consideration of the cutoff used for each platform, otherwise the comparison has no meaning [1,5]. In this section, we explore the challenge of selecting an “equivalent cutoff” for multiple assay platforms.

Each platform is designed to measure not the amount of enzyme but the velocity of the enzyme-catalyzed reaction in say a 3-mm punch of a DBS. All NBS assays are fixed-point rather than continuous, that is the velocity is taken as the amount of product formed at the end of the incubation period divided by the incubation time. This gives the velocity when the product versus time curve is linear [1]; we assume that this has been established for the various assay platforms being compared. In general, each assay platform may use a different substrate and buffer composition, thus even identical samples will give different velocities (say μmol/h/L). In principle, this difference can removed simply by dividing the velocity measured in any single DBS by the mean velocity measured across a large population of DBS, and reporting percent of mean activity for each newborn. As shown next, this scaling by normalization to the mean does not always lead to equivalent cutoff values for multiple assay platforms, and thus caution is advised.

What is measured in the assay is not the enzymatic product directly but an assay response (*AR*) that is proportional mostly to the amount of product, and is given by the general equation below.
*AR*(*t*) = *γ*[*P_EOI_*(*t*) + *P_OTE_*(*t*) + *P_NE_*(*t*) + *P*_IM_] + *δ*(3)
Here, *AR*(*t*) is the assay response at time *t*, *P_EOI_*(*t*) is the moles of product made by the enzyme of interest, *P_OTC_*(*t*) is the moles of product made by one or more off-target enzymes in the mixture (not all substrates are completely specific for the enzyme of interest), *P_NE_(t*) is the moles of product generated from substrate by all nonenzymatic processes, and *P*_IM_ is the moles of product present in the substrate as a contaminant (which is not a function of time). The constant *γ* links the total moles of product to *AR*. The parameter *δ* does not depend on the amount of product but is a contribution to *AR*(*t*) as it includes the assay response from the sample matrix and from the substrate itself and also any quenching of the assay response by the matrix (a negative contribution). For example, fluorimetric assays are subject to quenching by components of the matrix that absorb emitted light from the fluorophore. Mass spectrometry assays display matrix suppression of ionization of the product analyte. The latter is completely removed from consideration by use of a chemically identical, but isotopically substituted internal standard [6], whereas correction for fluorimetric quenching is difficult to achieve.

The desired enzymatic activity is given by
(4)$$\frac{P_{EOI}\left(t\right)}{t}$$


However, $P_{EOI}\left(t\right)$ is not readily obtained from *AR*(*t*) unless the terms unrelated to the enzyme of interest are known, and these additional terms can be significant. For example, in some fluorimetric assays of α-glucosidase for NBS of Pompe disease, *P_OTE_*(*t*) is ~20% of *P_EOI_*(*t*)*_,_* whereas for tandem mass spectrometry *P_OTE_*(*t*) is only ~3% of *P_EOI_*(*t*) [1]. For fluorimetric substrates using the 4-methylumbelliferone fluorophore, the substrate itself has significant fluorescence in the optical channel where the product is detected (a contribution to *δ*) [3]. Additionally, *P_OTE_*(*t*) and *δ* are likely to be sample-dependent, making it difficult to correct for all of these confounding factors using a single blank sample.

These considerations make it very challenging to pick a completely nonbiased, equivalent cutoff with which to compare the false negative and false positive rates for multiple assay platforms even when an identical set of DBS is used. These factors have been discussed in detail for NBS of Pompe disease and MPS-I, where the use of the same percent of mean enzymatic activity is probably satisfactory as an equivalent cutoff for MPS-I but not for Pompe disease [4].

The key result of the present study is that imprecision predominantly near the cutoff increases the rate of false positives and false negatives. NBS laboratories virtually always set their cutoff to be above the range of enzymatic activities measured in DBS from confirmed patients with the disease to avoid false negatives at the expensive of an increase in false positives. As explained above, the cutoffs may be different for different assay platforms even when expressed as a percentage of mean activity. Any reduction in the imprecision near the cutoff is expected to reduce the rate of false positive and false negatives and is thus a worthy goal in the development of a NBS platform. It is also clear that if platform 1 displays a lower imprecision at its cutoff compared to the imprecision of platform 2 at its cutoff, and, all other things being equal, platform 1 is expected to perform better than platform 2.

Consider the two black curves in Figure 3A, where one is for an assay with no-imprecision and the other is for an assay with imprecision. If the NBS laboratory wishes to hold the rate of false negatives to a constant value for both platforms by choosing the appropriate cutoff, one for each platform, the assay with the higher imprecision will give a higher rate of false positives (this can easily be seen by drawing a vertical line in Figure 3A to intersect the X-Axis at the desired false negative rate). Likewise, if one chooses cutoffs to keep the false positive rates the same on both platforms, the platform with the higher imprecision will result in a higher rate of false negatives.

## 9. Postanalysis Tools That Do Not Use Single Cutoffs

Postenzymatic activity measurement tools have been developed that try to generate a composite score based on measurement of more quantities than the enzymatic activity of a single enzyme. The most well-developed is the Collaborative Laboratory Integrated Reported (CLIR) suite of computational tools developed at the Mayo Clinic (reviewed in [4]). For example, measurement of the activities of six enzymes in a single DBS punch leads to more precise NBS than NBS based on the measurement of any single enzyme [7]. With six enzymatic activities in hand, it is possible to calculate all ratios of activities and to see if any of these ratios are in the range of the newborns with the disease in the training set versus in the range for pseudodeficiencies in the training set. The highest CLIR score to indicate that a newborn is likely to have the disease is when all the informative ratios are in the reference range for the DBS from the newborns with the disease in the training set. Thus, the CLIR analysis does not rely on a single cutoff for a single enzymatic activity. The biochemical basis for why certain enzymatic activity ratios are informative is usually not known, and it has been suggested that use of these ratios helps to normalize for differential white cell count in DBS [4]. It has already been noted above that sampling problems, such as the differential white cell count in DBS, apply equally to different assay platforms that measure the activity of lysosomal enzymes. Most importantly, postanalysis tools, including CLIR, cannot fix an assay imprecision problem. The only way to fix an assay imprecision problem is to modify the assay to reduce its imprecision or to carry out multiple independent measurements of the quantity in question so that a better estimate of its true value is obtained. The best option is probably to use CLIR to help account for sampling-dependent skewing (i.e., differential white cell counts for example) along with the assay with the lowest imprecision. It should also be noted that measurement errors propagate when ratios of measurements are used, again indicating that CLIR will not remove problems due to assay imprecision. Finally, this paper analyzes single-point assay methods, and thus one cannot directly compare single point methods to CLIR unless the statistical interdependence of the multiple assay measurements are known.

## 10. Concluding Remarks

In this article we have applied statistical analysis to NBS of lysosomal storage diseases as an example, but the formalism should be suitable for the NBS of other disorders. As found in several studies, there is considerable overlap between the enzymatic activities in newborns that do not have the disease and those that do. The problem is also confounded by the spectrum of severity of lysosomal storage diseases that present in early infancy and even as toddlers and adults. The data show that a single screen cutoff value will always be a trade-off between false negatives and positives, see for example [7].

There are many factors that lead to variation in the enzymatic activity measured in a 3-mm punch of a DBS measured across a population of newborns. In this paper we are interested in statistical methods to compare two different platforms for measuring the enzymatic activity in a DBS (for example tandem mass spectrometry versus fluorimetry). Thus, all factors that contribute to variation of enzymatic activity across the population of newborns other than assay imprecision are taken off the table in this comparison since they apply equally to both platforms. We establish that it is the assay imprecision near the screen cutoff that is most important for influencing the rate of false positives and negatives (or the rate of false positives if the rate of false negatives is chosen to be constant). In the case of curves of the type shown in Figure 3A that do not cross, one cannot erase the effect of a higher imprecision by adjusting the cutoff without paying a price. Said another way, if one insists on the same false negative rate for the two assay platforms, the one with the higher imprecision will necessarily lead to a higher rate of false positives.

Assay imprecision is measured by repeat analyses using a set of quality control DBS standards that are prepared to have identical levels of enzymatic activity. Since it is the assay imprecision near the screen cutoff that is most important, quality control standards with enzymatic activity near the cutoff are most useful. Thus, the BASE, MEDIUM, and HIGH quality control standards provided by the CDC are less useful than the LOW standard that gives an enzymatic activity closest to the screen cutoff typically used in NBS laboratories, see Figure 2. This logic leads to the suggestion that an improved set of quality control standards would be a set of 3 to 4 DBS that spans the screen cutoff in small increments (for example, 10, 15, and 20% of the population mean enzymatic activity). NBS laboratories typically measure the stability of the assay by repeat measurements of a Quality Control standard. It is advisable to use the quality control standard closest to the screen cutoff for this stability analysis. Quality control standards are also used to establish that the measured enzymatic activity varies linearly with the relative amount of enzyme in the DBS. However, it is not very important to show that that assay response is linear over enzymatic activity values far from the cutoff, but rather that it is linear in the region of the cutoff. In this paper we provide a method to incorporate the fact that the imprecision varies with assay value (Figure 2) in the analysis of false positives and negatives.

We have also demonstrated that there is a simple model for the underlying distribution that is independent of measurement platform and imprecision; the simple model requires only the mean and width of the distribution for both the normal and the diseased populations. With these four numbers, we can test the effects of assay imprecision. The Y-axes in the plots shown in Figure 3A,B indicate that we are able to extrapolate meaningfully and smoothly into a region of very small values in the PDF (or the false rates) and quantitatively test the effects of different types of imprecision. Curves of this sort are independent of cutoff, and can be used to inform clinicians of the choices made for a cutoff and prioritize the various missed individuals. It is also remarkable that small changes in the imprecision can have large effects on the rate (and hence the number) of false identifications. The false rates can be converted to actual numbers of patients if the prevalence of the disease is known. We suggest that the use of plots, such as those shown in Figure 3A,B, provide a way to be able to quantitatively compare various platforms. Lastly, NBS laboratories have various additional tests for patients near the cutoff, and we are providing the statistical analysis that justifies multiple tests for such patients.

## Figures and Tables

**Figure 1 IJNS-05-00017-f001:**
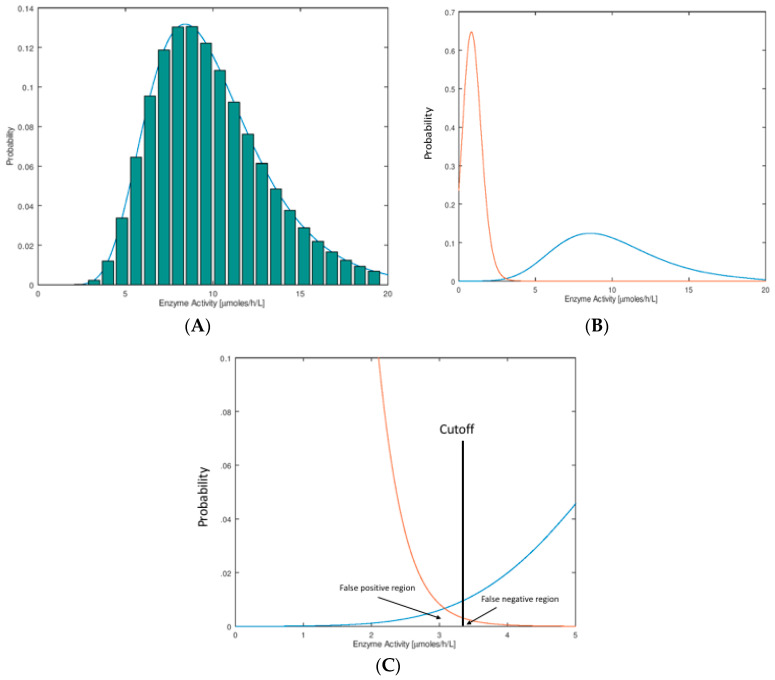
(**A**) Histogram distribution for enzymatic activity values. A collection of dried blood spots (DBS) from newborns is submitted to the enzymatic activity assay. The activity values are parsed into arbitrary bins, in this case of width 0.8 μmol/h/L. The height of each bin (bar in the plot) is the number of newborns displaying enzymatic activity in each bin divided by the total number of newborns. The smooth curve drawn on the histogram is a log-normal mathematical function fitted numerically to the histogram, by adjusting the three parameters, amplitude, mean, and width. (**B**) Shown are typical probability distribution functions (PDFs) for newborns that are healthy (blue) or confirmed to have the disease (orange). The orange curve intersects the origin and stops (difficult to see). The PDF for the healthy newborns is log-normal with a mean of 10 μmol/h/L and standard deviation 3.5 μmol/h/L. The PDF for the disease newborns is log-normal with mean 0.9 μmol/h/L and standard deviation 0.4 μmol/h/L (see Appendix A for the log-normal function). (**C**) Expansion of the plot in Figure 1B near the screen cutoff (vertical black line at 3.3 μmol/h/L, chosen as an example). The area under the healthy PDF to the left of the cutoff corresponds to false positives, and the area under the disease PDF to the right of the cutoff corresponds to false negatives. Note that there is no reason to put the cutoff at the point of intersection of the two curves. In this example, we choose a cutoff to the right of where the two curves intersect; this is typical of NBS laboratories where there is more desire to reduce false negatives to a minimum at the expense of an increase in false positives.

**Figure 2 IJNS-05-00017-f002:**
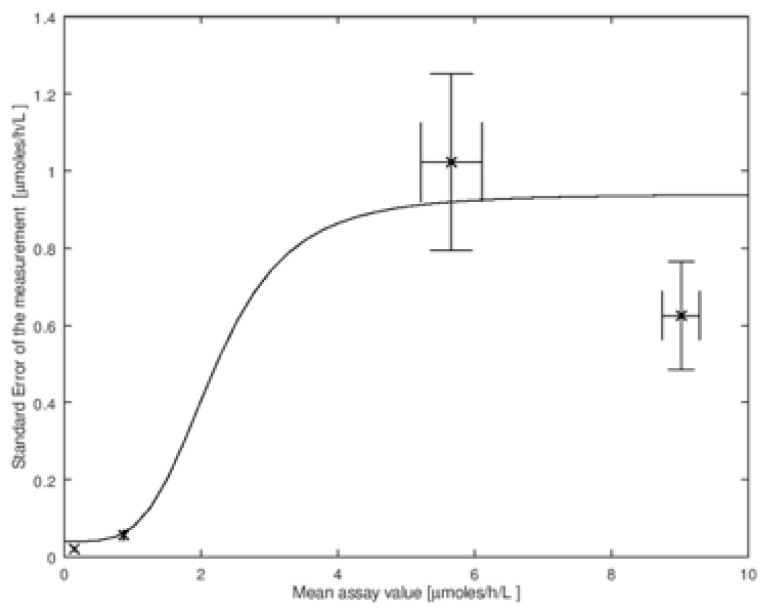
Plot of standard error (also called the standard deviation) of the measurement (*σ_M_*) versus the mean activity (*μ_M_*). This is for the tandem mass spectrometry assay of the GAA enzyme (relevant to Pompe disease) using the CDC Quality Control DBS. The data was obtained from the CDC (Set 2 2018 LSD FIA Certification) on their NBS web portal (https://www.cdc.gov/labstandards/nsqap_resources.html). The solid line is a model for the CDC data using the function: $c+a(x^{4}/\left(x^{4}+b^{4}\right)$ where *a*, *b*, and *c* are constants (this is a convenient function that well interpolates through the data points; it has no biological basis that we are aware of). Error bars in *σ_M_*, plotted on the *Y*-axis, are calculated as *σ_M_*/*N*^1/2^, where *N* is the number of punches analyzed by the CDC (*N* = 20). The error bars for *μ_M_*, plotted on the *X*-axis, are calculated as $t_{S}\sigma_{M}$/*N*^1/2^, at the 95% confidence interval, here *t_s_* is Student’s *t* value. These error equations are found in standard textbooks of statistics including the definition of Student’s *t* value, *t_s_*.

**Figure 3 IJNS-05-00017-f003:**
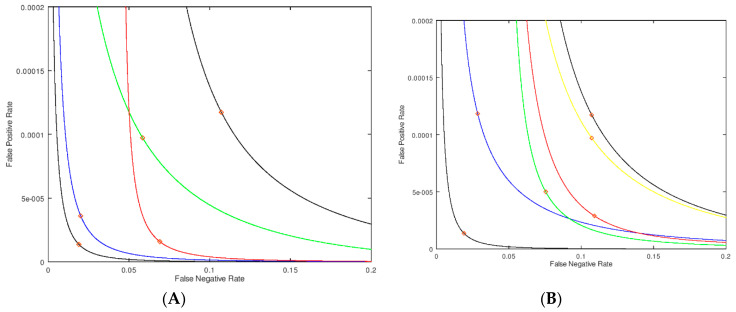
Parametric plots (modified received operating characteristic, ROC plots) where the false positive rate (fraction of healthy patients who are screen-positive) is plotted on the *Y*-axis, and the false negative rate (fraction of disease patients who are screen-negative) is plotted on the *X*-axis. The diamonds indicate the point on each curve for an assay value of 2.0 μmol/h/L (which could be chosen as the cutoff). The no-imprecision PDF is modeled by a log-normal distribution with a mean of 10.0 and a width of 3.8 μmol/h/L for the healthy population and a mean of 1.0 and width of 0.38 μmol/h/L for the patients with the disease (**A**). The black curve passing closest to the origin is for no-imprecision, and the other black curve is for uniform and high imprecision of *σ_M_* = 0.5 μmol/h/L for all values of enzymatic activity. The blue curve is for high imprecision of *σ_M_* = 0.5 μmol/h/L for enzymatic activities of 3 μmol/h/L or higher, and a negligible imprecision of *σ_M_* = 0.1 μmol/h/L for enzymatic activities below 3 μmol/h/L. The red curve has the high imprecision in the window of 0 to 1 μmol/h/L and negligible imprecision elsewhere. The green curve has high imprecision in the window of 1 to 3 μmol/h/L (passing through the cutoff) and negligible imprecision elsewhere. (**B**). Black curves as for Panel A. The blue curve has lowered imprecision (*σ_M_* = 0.25 μmol/h/L) in the window 0–2 μmol/h/L and high imprecision (*σ_M_* = 0.50 μmol/h/L) elsewhere. This region of lowered imprecision is slid to the right to give the green curve (*σ_M_* = 0.25 μmol/h/L in the window of 1–3 μmol/h/L), further to the right to give the red curve (*σ_M_* = 0.25 μmol/h/L in the window 2–4 μmol/h/L), and even further to the right to give the yellow curve (*σ_M_* = 0.25 μmol/h/L in the window 3-5 μmol/h/L).

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
