# Peer review of "The Importance of Assay Imprecision near the Screen Cutoff for Newborn Screening of Lysosomal Storage Diseases"

_2409-515X, 2019, doi:10.3390/ijns5020017_

Reviewer 1 Report

The authors provide a robust statistical analysis of the impact of assay imprecision on false positive and false negative rates. They provide a thorough review of the potential impact of multiple factors on these rates and successfully demonstrate that assay precision near an analyte cutoff plays a significant role in categorization of newborn’s screening result. The conclusion extrapolates the study results across all newborn screening methods, but the analysis focuses on an enzyme activity-based assay to demonstrate the statistical principles.  The study would benefit from a discussion of non-enzyme assays to support this extrapolation. This same suggestion applies to the comparison with post-analysis tools like CLIR.  While LSD tools rely on multiple enzyme measures, not all the system’s post-analytic tools have this requirement, and the study would benefit from a discussion of the non-enzyme tools in CLIR if the conclusion is that all newborn screening assays, regardless of analyte type, are impacted similarly by imprecision at the cutoff.  Finally, the authors suggest in lines 318 – 321 and 333 – 334 that retesting around a cutoff decreases imprecision without acknowledging that this is common practice in newborn screening laboratories.  While programs may not realize the statistical theory behind the practice or the impact on imprecision, the practice currently exists.

Author Response

Reivewer:  The study would benefit from a discussion of non-enzyme assays to support this extrapolation. This same suggestion applies to the comparison with post-analysis tools like CLIR.  While LSD tools rely on multiple enzyme measures, not all the system’s post-analytic tools have this requirement, and the study would benefit from a discussion of the non-enzyme tools in CLIR if the conclusion is that all newborn screening assays, regardless of analyte type, are impacted similarly by imprecision at the cutoff.  

Response:  This paper is not about CLIR.  This paper is about proving that the imprecision at the screen cutoff is the most important element of any screening method that uses a cutoff.  CLIR makes use of as many measured quantities as possible and that are known to be "informative".  By informative, it is meant that the quantity measured is at least partially different between samples from patients who have the disease  and those that do not (including those that give the measured quantity close to those measured from the disease group of samples).  If say 4 quantities are known to be informative, CLIR gives a composite score, i.e. the highest score suggesting a diseased newborn is obtained when all 4 quantities measured in any particular sample fits better with the disease range rather than with the non-disease range of values. So CLIR is not based on any single measurement with a single cutoff but on multiple measurements, but in the end CLIR is a cutoff method but the cutoff is for composite scores.  

As we have stated in our chapter, for each quantity that is measured in a CLIR method, the imprecision of the measurement near the interface between the values for the disease reference range and the healthy reference range is the most important assay element.  The same applies to each and every quantity being measured in the CLIR package. Thus, I think we have said enough about CLIR in our chapter.  Also we are only discussing single assay methods in our chapter and thus cannot directly compare to CLIR.

Reviewer:  Finally, the authors suggest in lines 318 – 321 and 333 – 334 that retesting around a cutoff decreases imprecision without acknowledging that this is common practice in newborn screening laboratories.  While programs may not realize the statistical theory behind the practice or the impact on imprecision, the practice currently exists.

Author:  Yes and no. The CDC provides QC samples for LSD NBS of the type Base, Low, Medium, and High.  The Low sample is closest to the typical screen cutoff but not really close enough in our opinion. The Low is always 5% of the High enzymatic activity, and High is typical of a random (healthy) newborn.  Yet the cutoff for say Pompe disease is typically around 15% of normal activity in many newborn screening labs.  It would be better if the CDC provides a QC sample closer to 15%.

Reviewer 2 Report

this is well presented, a relatively complex hypothesis is presented for the lay reader in statistics. it also stimulates the reader to seek to understand the principle of cut offs in screening. it is not a continuous and the student t test description captures ones attention. 

i would like to more about the role of imprecision and values close to the cut offs, the fact that this may increase false positives. More should be said the consequence of a false positive ezymatic tests on patient outcomes, the risks associated with false positive or false negative. by nature screening tests will present errors but minimising such errors by repeated tests may not be cost effective and repeated measures may not necessarily improve the value of the test. Errors associated with handling and storage of DBS, warrants more attention as this is important for day to day practice.

Author Response

Reviewer:  I would like to more about the role of imprecision and values close to the cut offs, the fact that this may increase false positives. More should be said the consequence of a false positive ezymatic tests on patient outcomes, the risks associated with false positive or false negative. by nature screening tests will present errors but minimising such errors by repeated tests may not be cost effective and repeated measures may not necessarily improve the value of the test.

Author:  I don't really understand what the first sentence from the reviewer means.   I think it is obvious to the readership of our chapter that false positives are to be minimized as they created anxiety for families and lead to more follow-up by the newborn screening laboratory and followup clinical centers.

Reviewer:  Errors associated with handling and storage of DBS, warrants more attention as this is important for day to day practice.

Author:  This article does not deal with handing of DBS and issues associated with that. Rather it deals with the downstream process, and leads to the conclusion that improving assay precision near the cutoff is the best you can do to improve the newborn screening process given a set of DBS samples.  Yes consistency of storage, etc.   are also good things presumably, but certainly not the topic of our chapter.